# Magnetic Properties of Manganese-Zinc Soft Ferrite Ceramic for High Frequency Applications

**DOI:** 10.3390/ma12193173

**Published:** 2019-09-27

**Authors:** Lucian-Gabriel Petrescu, Maria-Cătălina Petrescu, Valentin Ioniță, Emil Cazacu, Cătălin-Daniel Constantinescu

**Affiliations:** 1Department of Electrical Engineering, Faculty of Electrical Engineering, University “POLITEHNICA” of Bucharest, 313 Splaiul Independentei, RO-060042 Bucharest, Romania; 2IRCER-CNRS UMR 7315, University of Limoges, F-87068 Limoges, France

**Keywords:** soft magnetic, ferrite, ceramic, transformer core, demagnetization factor, losses

## Abstract

A soft magnetic MnZn-type ferrite is considered for high frequency applications. First, the morphological, structural, and chemical composition of the material are presented and discussed. Subsequently, by using a vibrating sample magnetometer (VSM), the hysteresis loops are recorded. The open magnetic circuit measurements are corrected by employing demagnetization factors, and by taking into consideration the local magnetic susceptibility. Finally, the hysteresis losses are estimated by the Steinmetz approach, and the results are compared with available commercial information provided by selected MnZn ferrite manufacturers. Such materials are representative in planar inductor and transformer cores due to their typically low losses at high frequency, i.e., up to several MHz, in low-to-medium power applications and providing high efficiency of up to 97%–99%.

## 1. Introduction

Planar transformers are an essential part of today’s high-end electronic devices due to their outstanding power density capabilities, responding to the increasing demand in creating various smaller-sized vs. cost-efficient types of converters [1]. In such units, size is directly related to switching frequency, with higher frequencies translating into smaller volume and also into higher efficiency if the right transformer design is chosen. Therefore, such devices are used to exact standards, with precise electrical characteristics such as capability, output, and aspect ratio. The design using printed circuit board (PCB) layers, which can be interleaved in such devices, offers lower values of the leakage inductance and consistent reducing of the windings’ losses [2,3].

Transformers made of the planar principle virtually eliminate all the shortcomings of old-fashioned wire wound types. In planar design, the windings are made of copper foil lead frames or printed circuit boards, i.e., flat copper spirals laminated on thin sheet dielectric substrates. These windings are then sandwiched along with appropriate insulators between large area, yet thin, state-of-the-art ferrite cores.

The most common soft magnetic materials used in planar transformer cores are ferrites [4,5,6,7], a class of ceramics with the typical formula M^2+^Fe^3+^O^2−^, where M^2+^ represents a divalent metal ion or a combination of divalent metal ions (A + B), such as manganese-zinc (MnZn). In comparison to other materials, ferrites reveal several advantages: high resistivity, wide range of operating frequency, low losses, high permeability, temperature stability, versatile of core shapes, and cost-efficiency. Nonetheless, a few disadvantages such as low saturation flux density, poor thermal conductivity, low tensile strength, and brittleness, may limit their use and reliability. The general compositional formula of MnZn ferrites is Mn_a_Zn_(1−a)_Fe_2_O_4_. Such compounds are prepared by using a ceramic process technology in four major steps: preparation of the powder from raw materials (organic or inorganic precursors), shaping the powder into cores, sintering, and finishing [8].

Faraday’s equation contains two important parameters from a magnetic point of view: the frequency of the signal *f*, and the saturation value of the magnetic flux density *B*_s_ [1,2,3]. The value of *B*_s_ is limited to the technologically breakthrough while the frequency of the signal might be increased by several MHz to insure an optimal value of applied voltage for the very small dimensions of the transformers. The most important advantages of such devices, when compared to classical transformers, are presented in literature [4,5,6,7], and they might be summarized as follows: low weight (15 g per 100 W), typical efficiency of ~97%–99%, small winding area, low profile that improves heat dissipation, and lower leakage inductance due to reduced/interleaved windings. 

Planar transformers find their applications in numerous high-end electronics, where a high switching frequency is required while the allowed space is reduced, e.g., in telecommunications, medical instruments, automotive vehicles, defence systems, power conversion, and interference suppression [7]. The core of any planar transformer is made of soft magnetic materials, which are needed to be easily magnetized, yet they also fulfill a few other requirements: high permeability, high saturation flux density, and low core losses. Several standard core geometries and shapes are commercially available, as defined by the industry and the international standards, e.g., classic EE and EI (E & I shaped cores), as well as ELP (E low profile), EFD (economic flat design), ETD (economic transformer design), PQ (power quality), and RM (rectangular modular) core types; the ER core geometry design (E core with round/circular windings) is the most common in commercial planar transformers [8,9,10,11,12]. The EE and EI cores have two open coil sides, and are used for voltage and current transformers due to their many benefits such as low noise, low magnetising current, fast assembly, and cost-efficiency. The EFD cores offer a significant advance in power transformer circuit miniaturization. ETD exhibits a round center post for minimum winding resistance, while dimensions are optimized for power transformer efficiency. The PQ cores feature round center legs with rather small cross-sections and are designed specifically for switched-mode power supplies, therefore providing an optimized ratio of volume to winding area and surface area. Originally, RM cores from Siemens have been essentially designed for two major applications, i.e., very low-loss, highly stable filter inductors and other resonance determining inductors, and in low-distortion broadband transmission at low signal modulation. Today, RM cores are increasingly required for power applications. Finally, ER cores are a cross between E cores and pot cores and represent an economical choice for planar transformers and inductors. A planar core consists of two flat pieces of magnetic material, one above and one below the coil, typically used with a flat coil that is part of a printed circuit board. The round center post offers minimal winding resistance, and better space utilization and shielding than with rectangular center leg planar cores. When compared with non-planar cores, ERs offer minimal height and better thermal performance. E/I combinations facilitate economical assembly, with typical applications in differential mode inductors and power transformers.

In this paper, we present results on a soft magnetic MnZn-type ferrite for high frequency applications. First, the morphological, structural, and chemical composition of the material are presented and discussed. Subsequently, the hysteresis loops are recorded and the open magnetic circuit measurements are corrected by employing demagnetization factors and by taking into consideration the local magnetic susceptibility. Finally, the hysteresis losses are estimated by the Steinmetz approach, and the results are compared with available commercial information provided by selected ferrite manufacturers.

## 2. Experimental 

### 2.1. Materials and Synthesis

Mn_a_Zn_(1−a)_Fe_2_O_4_ ferrites are produced by heating a mixture of finely-powdered nitrate-based salt precursors, and pressing them into a mold. Briefly, the inorganic chemicals that are employed without any purification are manganese nitrate–Mn(NO_3_)_2_ 4H_2_O, zinc nitrate–Zn(NO_3_)_2_ 6H_2_O, and iron nitrate–Fe(NO_3_)_3_·9H_2_O. The ratio between the cations is chosen with respect to the final desired composition of the ferrite. Here, we present and discuss results for Mn_a_Zn_(1−a)_ where a = 0.1 to 0.8, insisting on the Mn_0.75_Zn_0.25_Fe_2_O_4_ stoichiometry. Several techniques for obtaining such bulk ferrites are presented elsewhere in literature, via the mixing of various precursors and the subsequent sintering of the resulting powder [13,14,15,16]. For the specific magnetic measurements, bulk samples are cut into smaller, prismatic shapes (rectangular prisms) of various volumes, by using a diamond coated steel wire (Well, Precision Wire Type A6, 250 µm diameter) on a setup from Well/Walter EBNER (“Precision Horizontal Diamond Wire Saw”, model 3032-4).

### 2.2. Chemical, Morphological & Structural Analysis, and Magnetic Properties

Scanning electron microscopy (SEM) has been used to assess information about the samples surface topography and composition in a freshly broken/exposed surface. A field emission gun (FEG-SEM) setup from FEI (model Quanta FEG 450, Limoges, France) was equipped with a secondary electron detector (model SE R580, Limoges, France), as the number of secondary electrons emitted by atoms excited by the electron beam that can be detected depends, among other things, on specimen topography. Energy-dispersive X-ray spectroscopy (EDS), as an effective technique that enables one to swiftly perform a qualitative and quantitative inspection of the elemental composition of samples, has been employed as the analytical technique of choice for the elemental analysis and chemical characterization of the MnZn samples. An EDS device from Oxford Instruments (model X-Max 150, Limoges, France) equipped with a very large area (150 mm^2^) silicon drift detector (SDD) was used, installed in situ on the FEG-SEM setup. The AZtec analysis software from Oxford Instruments plc, UK was employed for analysis and data processing. X-ray photoelectron spectroscopy (XPS) was used as a second choice corroborative investigation technique to probe the elemental composition at the parts per thousand range, the empirical formula, and chemical and electronic state of the elements within the material. The samples were placed in an ultrahigh vacuum chamber (10^−9^ mbar) of the spectrometer from Kratos (model “AXIS Ultra DLD”, Limoges, France) and irradiated by a monochromatic x-ray beam (Al Kα, 1486.6 eV), while the experimental data were acquired by using by a 128 channel detector (Delay Line system) and “Vision 2 Software,” and were further analyzed with the CasaXPS software in order to extract the composition. X-ray diffraction (XRD) was employed to reveal structural information, crystal structure, crystallite size, strain, and preferred orientation of the MnZn ferrite. XRD patterns have been recorded in air and at room temperature on theta/2theta geometry by using a Cu_Kα_ x-ray source (λ = 0.154 nm) on a setup from Bruker (model D8 Advance), equipped with a focusing Ka1 mono filter and a LynxEye (model A17-B60) detector (samples in rotation during x-ray scanning at 30 rpm). The “DIFFRAC.SUITE” EVA software from Bruker (Limoges, France) was used for XRD data analysis, making use of the reference databases ICDD PDF2/PDF4+/PDF4 Minerals for phase identification and accurate quantitative phase analysis were used in particular to track minor phases. Sample density was determined by helium gas pycnometry on a setup from Micromeritics (model AccuPyc II 1340). The dimensions of the cut samples varied in order to fit the measurement area of the magnetic measurement equipment, a vibrating sample magnetometer (VSM) from LakeShore (model 7304), and made use of the demagnetization procedure. The VSM has high sensibility (±0.05% per day) and excellent reproducibility in magnetic fields up to 3.4 T.

## 3. Results and Discussion

Most ferrites are spinels [13] with the formula A_x_B_y_Fe_2_O_4_, where A and B represent various “2+” metal cations and “*x*” and “*y*” can be chosen in various proportions. The spinel structure has been found to be significantly influenced by the preparation of the ferrite ceramic. To fit a selected application, it is important to control the global composition of the ferrite with respect to the *Fe*_2_*O*_3_ content (mol %). Mn_a_Zn_(1−a)_Fe_2_O_4_, as the general compositional formula of MnZn-type ferrites, requires an adequate control of the precursor quantity and type in order to fit the desired stoichiometry. Spinel ferrites usually adopt a crystalline structure consisting of cubic close-packed (*fcc*) oxides (O^2−^), with A^2+^ and B^2+^ cations occupying one eighth of the tetrahedral sites or holes and Fe^3+^ cations occupying half of the octahedral sites. For example, the naturally occurring mineral “Franklinite” is an oxide mineral belonging to the normal spinel subgroup’s iron series with the formula Zn^2+^Fe^3+^_2_O_4_. However, ferrite crystals may not only adopt the ordinary spinel structure, but also an inverse spinel structure. Therefore, it is also possible to have mixed structure spinel ferrites with the formula (M_2+1−δ_Fe_3+δ_)_1_ (M_2+δ_Fe_3+2−δ_)O_4_, where δ represents inversion degree [17]—see Table 1. To illustrate, another interesting mineral is “Jacobsite,” a manganese iron oxide mineral. It is also in the spinel group and forms a solid solution series with magnetite. The chemical formula is MnFe₂O₄, or with the oxidation states and substitutions, it is (Mn^2+^,Fe^2+^,Mg)(Fe^3+^,Mn^3+^)_2_O_4_. Furthermore, other manganoan derived minerals and compounds, i.e., structures containing divalent manganese, as well as zincian, i.e., containing zinc, are also known. Yet, divalent iron and/or manganese may commonly accompany zinc, and trivalent manganese may substitute for some ferric iron. If one eighth of the tetrahedral holes are taken by the Fe^2+^ cation, then one fourth of the octahedral sites are occupied by one of the A or B cations, in our case Mn^3+^, and the other one fourth by the Fe^3+^ cation; this is the inverse spinel structure [17]. Therefore, we consider here different compositions and structures. EDS, XPS, SEM, and XRD data are presented, corroborated, and discussed with respect to the final usable composition of the material. Several similar structures and compositions are also presented and discussed elsewhere [16,17,18,19,20].

### 3.1. Structure and Magnetic Domains Size

As previously stated, with respect to a specific application, it is important to control the composition (mol %) for Fe_2_O_3_, MnO, and ZnO, respectively. This aspect will ensure an optimum performance for saturation flux density (*B*_s_), low losses (*P*_Fe_), and initial permeability (*μ*_i_) (Figure 1), and one can identify the Curie temperature (*T*_c_) lines at 100 °C and 250 °C, respectively. In an attempt to enhance magnetic flux density at around 100 °C, we examined several possibilities, e.g., the influence of chemical structure. Several compositions have been considered, and are presented in Figure 1 and Table 1, respectively. 

The aspect and microstructure of the MnZn ferrite is presented in Figure 2 in a freshly exposed surface as assessed by SEM. One may observe that the structure of the compound is granular ceramic microcrystalline with grains typically below 20 µm in diameter, homogeneous, and highly compact. The average grain diameter size has been limited to a maximum of 20 µm, otherwise pores appear in its grain boundary as well as in the grain itself, and this lower the initial permeability by the locking of the wall. A drawback of MnZn ferrites is their saturation magnetic flux density, which is lower than those in metallic soft materials, as it will later be discussed. This means that a larger volume of ferrite core is required to produce the same amount of magnetic flux as metallic cores produce. Furthermore, the Curie temperature of the MnZn ferrite is also lower, being typically less than 250 °C. Magnetic flux density decreases as temperature increases, and it vanishes at the Curie temperature. This means that magnetic flux density is lower at the operating temperature of transformers, from 80 to 100 °C, than at the room temperature.

EDS investigation, a simple yet rapid and accurate analytical technique used for the elemental analysis or chemical characterization of a sample, reveals the chemical composition of the MnZn-type ferrite samples after the application of quantitative correction procedures, as presented in Figure 3. Following the results for the atomic and weight percentage, the chemical structure of the compound is found to be Mn_0.75_Zn_0.25_Fe_2_O_4_, corresponding to the desired choice with respect to its potential application in planar transformers [1,2,4,5,6,7]. However, as the accuracy of this quantitative analysis of sample composition is affected by various factors, to accurately evaluate the empirical formula, chemical, and electronic state of the elements within the material, XPS analysis was also employed. Under the effect of Ar^+^ ions bombardment, electrons from the sample are ripped off and are collected by a 128 channel detector for spectroscopy and imaging, and further analyzed by a 180° hemispherical analyzer with a 165 mm radius. A first measurement is achieved, and then another ionic etching is performed by Ar^+^ ions with an acceleration voltage of 2 kV for one minute over 3 × 3 mm² in order to overcome the contamination of the analyzed surface. Data are acquired by using Vision 2 Software and analyzed by using CasaXPS software to extract it. The composition of the films is then adjusted (energy correction of the spectra) for the ferrite samples by its comparison to the peak C 1s of the carbon placed at 285 eV.

The results have been further corroborated with the XRD data. The crystalline structure of the MnZn ferrite samples is presented in Table 1, and peaks are indexed in Figure 4. The peaks detected in the diffractograms reveal a cubic structure and confirm the lack of formation of any secondary or minor phase. One may observe that, by analyzing the peaks in the respective difractograms with respect to the XRD databases, the chosen compound is best fitted by structures ranging from Mn_0.6_Zn_0.4_Fe_2_O_4_ (pdf #01-074-2401) to Mn_0.80_Zn_0.18_Fe_2.02_O_4_ (pdf #04-016-2782). The (Mn_0.612_Zn_0.288_Fe_0.1_)_1_(Fe_9.835_Mn_0.0165_)_2_O_4_ structure (pdf #01-085-1202) also fits the XRD peaks well, revealing that it is rather possible to have a mixed structure spinel ferrite. On the other hand, samples sintered at 1300 °C exhibit more spinel formation than the others [14]. The samples are polyoriented and polycrystalline.

Finally, the samples are cut to suit the magnetic measurements. Thus, the rectangular prism sample volume is 5.2974 mm^3^ with 2a = 1.737 mm, 2b = 1.724 mm, 2c = 1.769 mm (Figure 4, inserted image), and a mass of 0.0225 g. The mass density of the sample is 4.780 kg/m^3^, as determined by helium gas pycnometry, similar to [15].

### 3.2. Magnetic Characteristics and Correction Fact

The VSM-recorded magnetization vs. magnetic field strength loops without demagnetization correction are presented in Figure 5. Measured hysteresis loops have been obtained for the investigated samples, considering the applied field in the plan of the sample (blue) and perpendicular to the sample plan (red), respectively. The samples have been analyzed in two perpendicular directions to verify any existing anisotropy in the ferrite. Nevertheless, as one can observe, both curves almost overlap. Traditionally, the characterization of soft magnetic materials is performed by using closed magnetic circuit equipment, such as the Epstein frame or the single sheet tester [11]. However, when analyzing small samples such as in our case, they have no proper size and dimensions to be measured with by this method. Therefore, the VSM could be a solution, but the measurement in an open magnetic circuit may lead to inaccurate results due to the demagnetization field and the influence of the electromagnetic poles [11]. 

Several scientific papers are discussing the issue of the demagnetization field [21,22,23,24]. By knowing the shape of the sample, the demagnetization factors may be calculated, but it is important to also consider the local susceptibility of the material [25]. The literature considers two types of demagnetization factors: magnetometric, *N*_m_, and fluxmetric, *N*_f_, respectively.

In our further study, we will refer to the magnetometric demagnetization factor. The interior magnetic field *H*_ef_ of a homogeneous sample placed in a uniform applied field *H*_a_ may be written as:
(1)$$H_{\mathrm{ef}}=H_{a}-N_{m}\times M=H_{a}-N_{m}\times \chi \times H_{\mathrm{ef}}$$
where *χ* is the magnetic susceptibility of the sample. Thus, we can rearrange the formula:
(2)$$H_{\mathrm{ef}}=\frac{H_{a}}{1+N_{m}(\chi )\times \chi }$$
this being the value of the corrected field that is to be applied for the measurements. Generally, the dependency of the demagnetization factor on the magnetic susceptibility is indicated tabularly for certain values of this parameter [22,23,24,25], or by using FEM software such as in [12]. Here, a fitting procedure from MATLAB^©^ was used to obtain the local value of the demagnetization factor starting from the tabular values presented before. Several fitting functions have been used, but the best results were obtained by using the following hyperbolic evolution (the relative errors between the fitting values and those from the tables presented in literature have also been considered):
(3)$$N_{m}(\chi )=\frac{N_{m}(0)-N_{m}(\infty )}{{\left(1+\chi \right)}^{n}}+N_{m}(\infty )$$
where N_m_(0) is the value of the magnetometer demagnetizing factor for *χ* = 0, N_m_(∞), the same factor for *χ* = ∞, and n is a fitting parameter that offers the best result for the value of 0.67.

By using Equation (4), the corrected magnetic loops are presented in Figure 6, a comparison between the hysteresis loops with and without the applied magnetization factor, considering only the low magnetic field area. The iterative procedure was applied until the values of the magnetic susceptibility in a low field were close to the commercial data available on this ferrite composition. The maximum value obtained here is 5.099, and the manufacturers specify the value of the magnetic permeability between 3.000 and 5.500 for most of the ferrite types [26].

### 3.3. Losses Estimation in High Frequency

An important part of this study is to determinate the variation of the magnetic losses with the frequency because the planar transformers are working in range of 200–700 kHz. For this reason, the first step was to transform the measurement from the *M*(*H*) system to *B*(*H*) (Figure 7). The integration over a loop offers us the density of the magnetic energy. Considering the volume of the sample, the magnetic energy is 0.0693 mJ.

To estimate the power losses in this material, the Steinmetz approach [27,28,29] was used by adopting the following Equation:
(4)$$P_{\mathrm{hyst}}=f\times C_{m}\times B_{s}^{\alpha }$$
with *C*_m_ is the Steinmetz hysteresis coefficient, and α is a losses coefficient. Sakaki [30] considered the coefficient α = 1.64 for magnetic flux densities lower than 1 T (as in our case, for MnZn ferrites). The Steinmetz hysteresis coefficient might determinate the static value of the magnetic energy:
(5)$$W_{hyst}=C_{m}\times B_{s}^{\alpha },$$
and the power losses can be computed for different values of the magnetic flux density. The core manufacturer [6] indicated the total power losses for different values of the working frequency and different values of the peak magnetic flux densities. By using the Steinmetz approach, the variation of the hysteresis losses with the frequency for different values of the flux density is shown in Figure 8. Such results might be used in dissimilar situations by considering some of the hysteresis models of various materials [31,32,33,34,35]. To conclude, MnZn ferrites are representative in planar inductor and transformer cores due to their typically low losses at high frequency. Appropriate in low-to-medium power applications and providing high efficiency of up to 97%–99%, it was shown that the maximum transmissible power is reached at a switching frequency around 2–4 MHz and an operating temperature of 100 °C. The outstanding properties of this material will enable considerably more compact power supplies to be designed in future, therefore contributing to significantly greater energy savings.

## 4. Conclusions

Mn_a_Zn_(1−a)_Fe_2_O_4_ ferrites are produced by using a mixture of finely-powdered nitrogen-based salt precursors pressed into a mold, and subsequently sintering the resulting powder. The bulk samples are cut into smaller prismatic shapes (rectangular prisms) of various volumes, by using a diamond coated steel wire. The magnetic characterization of the MnZn type ferrite provides one with valuable information, as the open sample results have been corrected with demagnetization factors, which consider the punctual susceptibility of the material. The novelty of this work consists of the estimation of these factors by an iterative procedure, i.e., by using an exponential fitting procedure. The corrected values are presented and discussed and fit manufacturers’ data, which are provided as a range of values. The last part of the study focuses on estimating the magnetic losses in high frequencies for this ferrite by using the Steinmetz approach, i.e., the hysteresis losses are determined for the MnZn ferrite. All these properties make MnZn-type ferrites ideal for applications in a variety of high frequency transformers, adjustable inductors, wide band transformers, and high frequency circuits from 10 kHz to 50 MHz, while the impedance of these cores make them ideal for inductors up to 100 MHz. Furthermore, special ferrite-particle based paints may be used as absorbing materials in the microwave frequency range (1–300 GHz) as a component of radar-absorbing materials or coatings in stealth aircrafts, and in the absorption tile lining in the rooms used for electromagnetic compatibility measurements.

## Figures and Tables

**Figure 1 materials-12-03173-f001:**
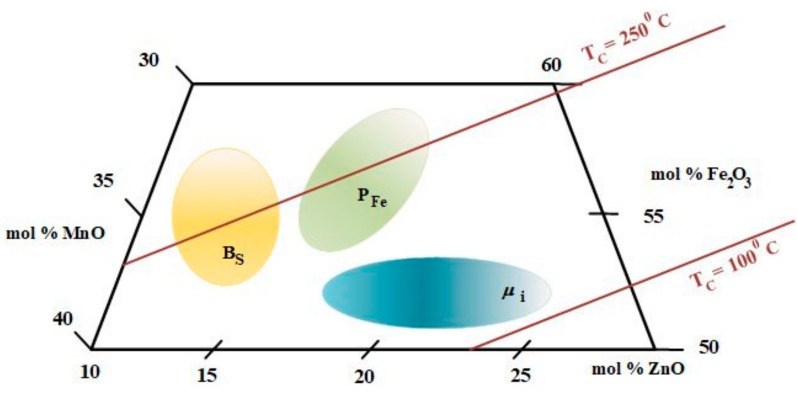
Composition diagram for MnZn ferrites, adapted from [11].

**Figure 2 materials-12-03173-f002:**
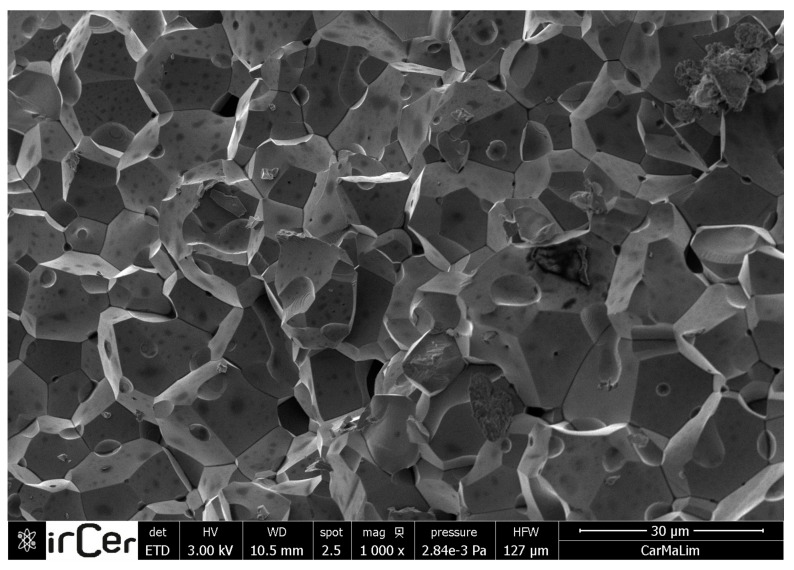
Scanning electron microscopy (SEM) of the MnZn ferrite revealing the granular microstructure, with grains typically below 20 µm in diameter.

**Figure 3 materials-12-03173-f003:**
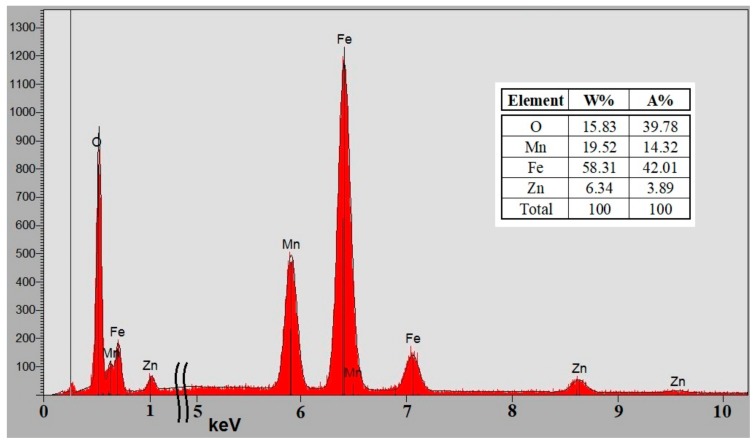
Energy-dispersive X-ray spectroscopy (EDS) of the MnZn ferrite revealing the chemical composition of the material.

**Figure 4 materials-12-03173-f004:**
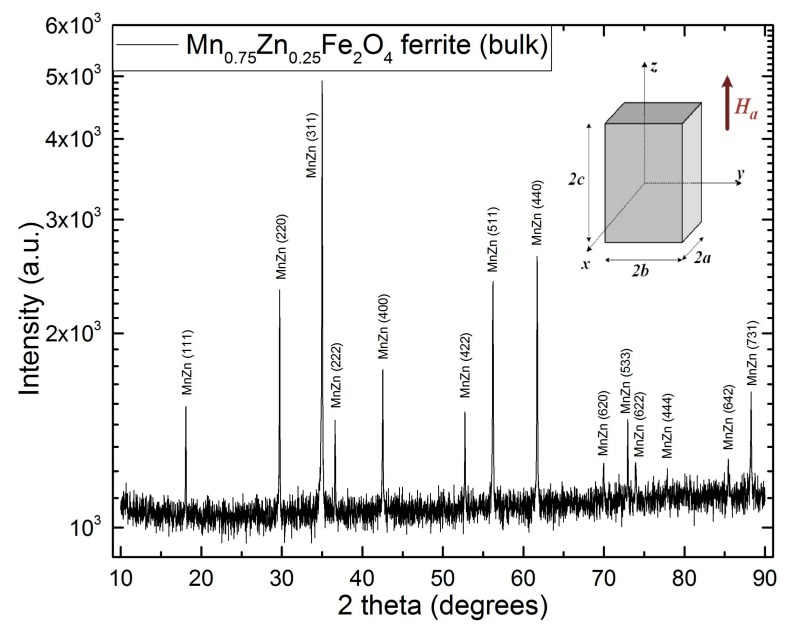
X-ray diffractogram (XRD) of the MnZn ferrite revealing the poly-oriented and polycrystalline structure; insert image: dimension of the cut samples to be measured by the vibrating sample magnetometry (VSM) technique, and direction of the applied magnetic field.

**Figure 5 materials-12-03173-f005:**
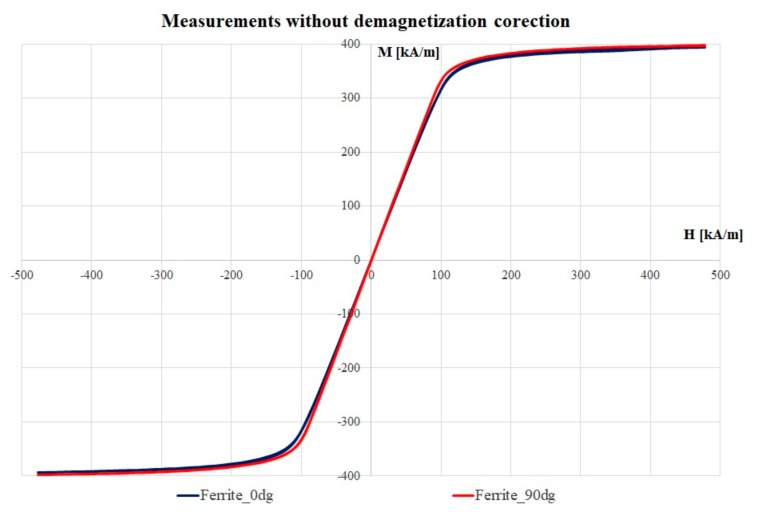
Measured hysteresis loops obtained for the investigated sample, considering the applied field in the plan of the sample (blue) and perpendicular to the sample plan (red), respectively.

**Figure 6 materials-12-03173-f006:**
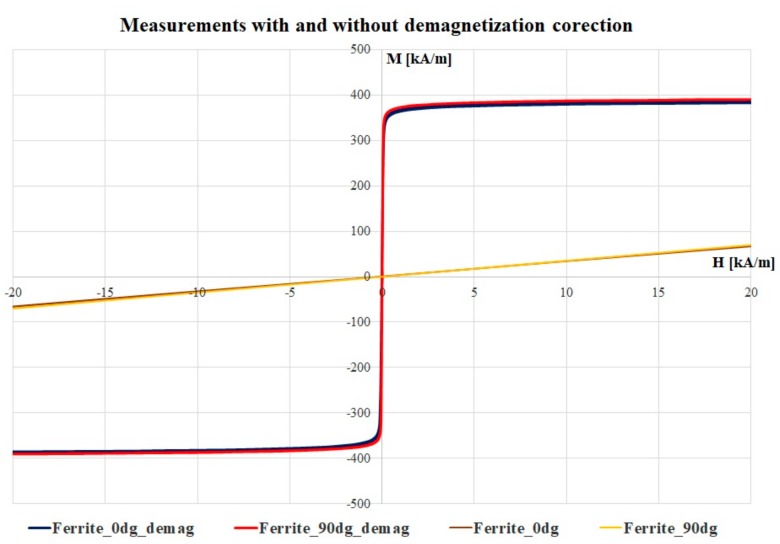
Comparison between the hysteresis loops with and without the applied magnetization factor, considering only the low magnetic field area.

**Figure 7 materials-12-03173-f007:**
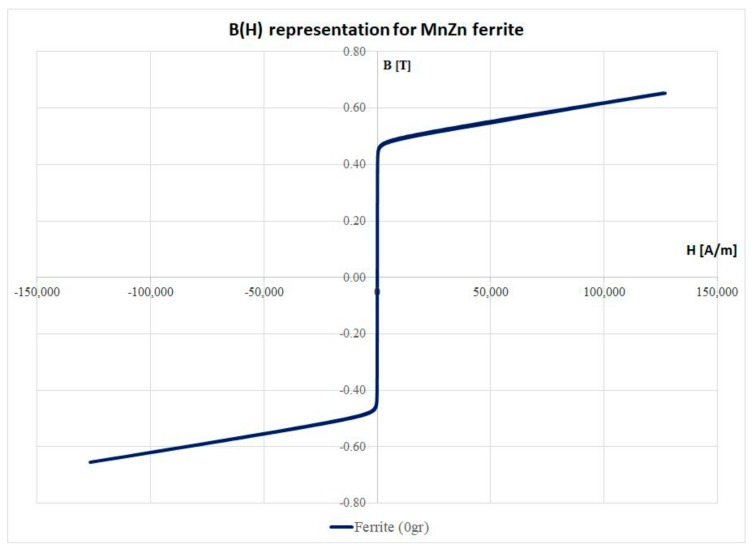
B(H) representation for the investigated sample.

**Figure 8 materials-12-03173-f008:**
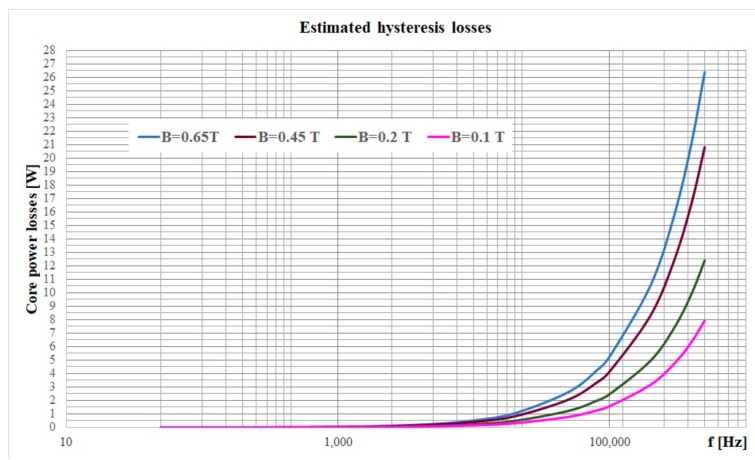
The estimated hysteresis losses for different flux densities of the investigated material.

**Table 1 materials-12-03173-t001:** Chemical composition, crystalline structure, and lattice parameter for the Mn_a_Zn_(1−a)_ ferrite compositions, where a = 0.1 to 0.8, and the corresponding XRD powder diffraction database (pdf) files.

File (pdf #)	Chemical Stoichiometry	CrystallineStructure	Observation
04-016-2782	Mn_0.80_Zn_0.18_Fe_2.02_O_4_	Cubic, a = 8.5	Manganese zinc iron oxide
04-002-0493	Mn_0.7_Zn_0.3_Fe_2_O_4_	Cubic, a = 8.5	Zinc manganese iron oxide
04-002-0227	Mn_0.69_Zn_0.31_Fe_2_O_4_	Cubic, a = 8.474	Zinc manganese iron oxide
01-085-1202	(Mn_0.612_Zn_0.288_Fe_0.1_)_1_(Fe_9.835_Mn_0.0165_)_2_O_4_	Cubic, a = 8.4915	Manganese zinc iron oxide
01-074-2401	Mn_0.6_Zn_0.4_Fe_2_O_4_	Cubic, a = 8.4975	Jacobsit, Zincian
00-068-0293	Mn_0.5_Zn_0.5_Fe_2_O_4_	Cubic, a = 8.4344	Manganese zinc ferrite, Manganese zinc iron oxide
01-080-6581	(Mn_0.5_Zn_0.5_)Fe_2_O_4_	Cubic, a = 8.462	Franklinite, Manganoan
01-074-2400	Mn_0.4_Zn_0.6_Fe_2_O_4_	Cubic, a = 8.4794	Franklinite, Manganoan
01-074-2399	Mn_0.2_Zn_0.8_Fe_2_O_4_	Cubic, a = 8.4616	Franklinite, Manganoan
01-074-2398	Mn_0.1_Zn_0.9_Fe_2_O_4_	Cubic, a = 8.453	Franklinite, Manganoan

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
