# Peer review of "Magnetic Properties of Manganese-Zinc Soft Ferrite Ceramic for High Frequency Applications"

_materials, 2019, doi:10.3390/ma12193173_

Round 1

Reviewer 1 Report

Experimental part: requires detailed descriptions as to the synthesis methods used and the reagents used.
Please provide details on the synthesis carried out by these two methods. Was the goal to receive new materials or to restore those ferrites referred to by the authors in 2.1. materials and synthesis, cited articles [13-15].
What was new about ??

2.2 Chemical, morphological....

EDX and XPS methods were used to characterize the obtained materials. However, these are not methods that can be used to perform elementary chemical analysis (determining the chemical composition of the resulting particles). For this purpose should be done eg. ICP-OES.

Row 107: no reference to the latest publications on ferrites for example:
https://www.mdpi.com/1996-1944/12/7/1048
https://doi.org/10.3390/ma12111871
https://www.mdpi.com/1996-1944/12/11/1871 etc....

What was the composition of Mn(a)Zn(1-a)Fe2O4, whose XRD results are presented in Fig. 4. Please, compare the results of the experiment with the standard MnZn ferrite spectrum on one graph.

Please clarify the description of VSM measurements.

English language and style are fine/minor spell check required.

Author Response

Please find enclosed the revised form (R1) of the manuscript. Your suggested corrections and corrections of our own consideration, respectively, are visible throughout the text in red colour. Next, our response to the Reviewer(s).

Reviewer 2 Report

Please see the attached referee report.

Author Response

Dear Reviewer #2,

In attach, please find enclosed our response letter. Your suggested corrections and corrections of our own consideration, respectively, have been made.

We appreciate very much that you re-consider this paper for peer-review, for subsequent publication in the journal MDPI “Materials”.

Thank you!

Sincerely,
The Authors

Round 2

Reviewer 1 Report

No comments

Author Response

Dear Reviewer #1,

Thank you for your time in evaluating our work. Please find enclosed the revised version (R2) of the manuscript. Your suggested corrections and corrections of our own consideration, respectively, are visible throughout the text in red colour.

Sincerely yours,
The Authors

Reviewer 2 Report

Please see the attached referee report.

Author Response

Dear Reviewer #2,

Thank you for your time in evaluating our work. Please find enclosed the revised version (R2) of the manuscript. Your suggested corrections and corrections of our own consideration, respectively, are visible throughout the text in red colour.

Sincerely yours,
The Authors

Round 3

Reviewer 2 Report

The manuscript can now be accepted for publication.